# MRAP2 Inhibits β-Arrestin-2 Recruitment to the Prokineticin Receptor 2

**Roberta Lattanzi** [1], **Ida Casella** [2], **Maria Rosaria Fullone** [3], **Daniela Maftei** [1], **Martina Vincenzi** [1] and **Rossella Miele** [3,*]

[1] Department of Physiology and Pharmacology "Vittorio Erspamer", Sapienza University of Rome, Piazzale Aldo Moro 5, 00185 Rome, Italy; roberta.lattanzi@uniroma1.it (R.L.); dani3la_maftei@yahoo.com (D.M.); martina.vincenzi@uniroma1.it (M.V.)

[2] National Centre for Drug Research and Evaluation, Istituto Superiore di Sanità, 00161 Rome, Italy; ida.casella@iss.it

[3] Department of Biochemical Sciences "A. Rossi Fanelli", Sapienza University of Rome, Piazzale Aldo Moro 5, 00185 Rome, Italy; mariarosaria.fullone@uniroma1.it

[*] Correspondence: rossella.miele@uniroma1.it

**Abstract:** Melanocortin receptor accessory protein 2 (MRAP2) is a membrane protein that binds multiple G protein-coupled receptors (GPCRs) involved in the control of energy homeostasis, including prokineticin receptors. These GPCRs are expressed both centrally and peripherally, and their endogenous ligands are prokineticin 1 (PK1) and prokineticin 2 (PK2). PKRs couple all G-protein subtypes, such as Gαq/11, Gαs, and Gαi, and recruit β-arrestins upon PK2 stimulation, although the interaction between PKR2 and β-arrestins does not trigger receptor internalisation. MRAP2 inhibits the anorexigenic effect of PK2 by binding PKR1 and PKR2. The aim of this work was to elucidate the role of MRAP2 in modulating PKR2-induced β-arrestin-2 recruitment and β-arrestin-mediated signalling. This study could allow the identification of new specific targets for potential new drugs useful for the treatment of the various pathologies correlated with prokineticin, in particular, obesity.

**Keywords:** G-protein coupled receptors; prokineticin receptors; melanocortin receptor accessory protein 2; β-arrestin-2





## 1. Introduction

GPCR-mediated signalling begins with the binding of the ligand to the receptors, followed by a change in receptor conformation and activation of the heterotrimeric G protein. G protein signalling is usually terminated by the recruitment of β-arrestins (β-arrestin-1 or β-arrestin-2), which sterically compete for binding with G proteins and desensitise GPCRs. β-arrestins also act as adaptors for clathrin to induce internalisation of the receptor and as a scaffold to promote signalling that induces mitogen-activated protein kinases (MAPKs) [1–3].

Prokineticin receptors (PKRs) are GPCRs that are expressed both centrally and peripherally and whose endogenous ligands are prokineticin 1 (PK1) and prokineticin 2 (PK2) [4]. They are able to couple all G-protein subtypes, such as Gαs, Gαq/11, and Gαi, and mediate several signalling pathways such as that of cyclic adenosine monophosphate (cAMP), protein kinase C, extracellular signal-regulated kinases (ERK), phosphoinositide 3-kinase, STAT3, or protein kinase B [5–12]. In addition, PKRs recruit β-arrestins upon PK2 stimulation, although the interaction between prokineticin receptor 2 (PKR2) and β-arrestins does not trigger receptor internalisation [13,14]. The diversity of G proteins coupled to PKRs and the alternative splicing isoforms of both PK2 [15,16] and PKR2 [17], the dimerisation of PKR2, and the interaction with accessory proteins [5] contribute to modulate receptor-specific functional properties in different tissues (Scheme 1). PK2 has

numerous physiological functions, including the regulation of neurogenesis, angiogenesis, pain threshold, mood, circadian rhythm, and food intake [5,6,18–20].

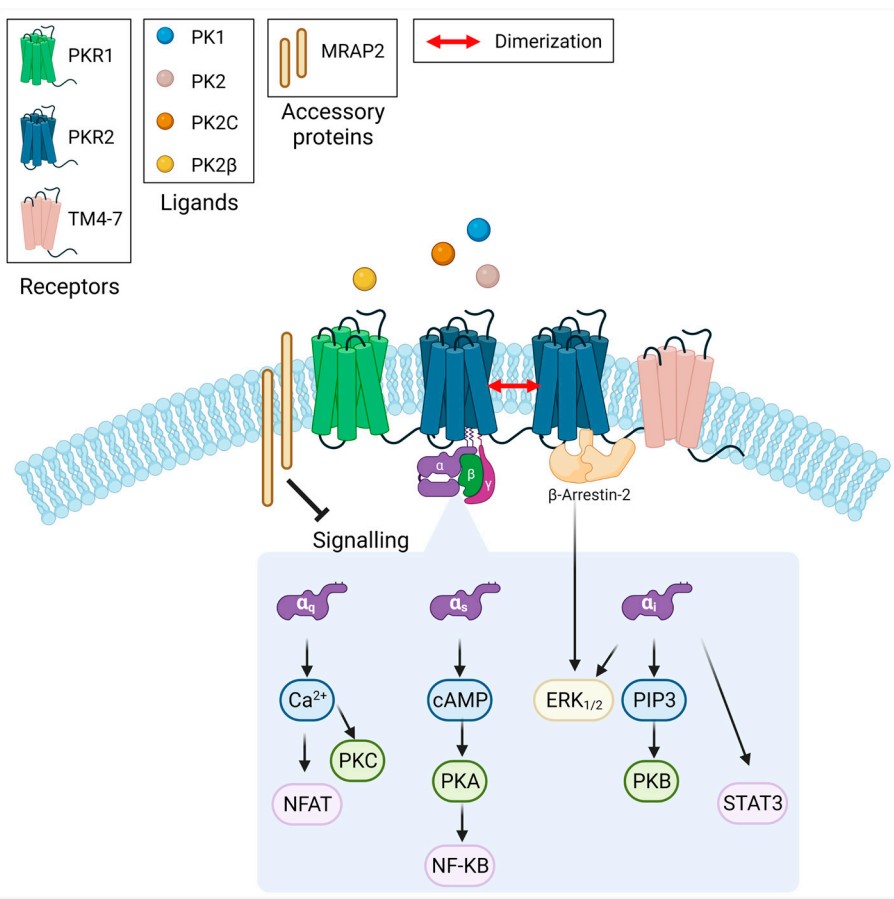

**Scheme 1.** Prokineticins bind prokineticin receptors and activate multiple signal transduction pathways.

Recently, it has been demonstrated that PK2, binding to prokineticin receptor 1 (PKR1), acts as an adipokine and reduces food intake and adipose tissue proliferation via the hypothalamic arcuate nucleus (ARC)-melanocortin system [18–20]. In mice, global ablation of PK2 or PKR1 leads to obesity [6]. The activation of PKRs is modulated by melanocortin receptor accessory protein 2 (MRAP2), a membrane protein that binds several G protein-coupled receptors (GPCRs) involved in the control of energy homeostasis [21–28]. The anorexigenic effect of PK2 through the binding of PKR1 is enhanced in MRAP2 knock-out (KO) mice compared to wild-type (WT) mice, demonstrating the inhibitory role of MRAP2 on PKR1 in vivo [29]. It has also been shown that PKR2 in the amygdala mediates the anorexigenic effect of PK2 and that variations in MRAP2 expression in PKR2 neurons alter PK2 activity [30]. The aim of this work was to elucidate the role of MRAP2 in modulating PKR2-induced β-arrestin-2 recruitment and β-arrestin-mediated signalling. We used BRET to analyse the interaction of PKR2 with β-arrestin-2 in the presence of MRAP2 and the direct interaction of MRAP2 with β-arrestin-2 has been evaluated by using biochemical techniques. The results demonstrate that MRAP2, acting as an allosteric regulator, modulates PKR2-induced β-arrestin-2 recruitment. This study may allow for the identification of allosteric sites as new specific targets for potential drugs useful for the treatment of the different pathologies correlated with prokineticin, especially obesity [5,31–37].

## 2. Materials and Methods

### 2.1. Drugs and Reagents

Dulbecco's Modified Eagle Medium with F-12 supplement (DMEM/F-12) media, foetal bovine serum (FBS), phosphatases inhibitor cocktails, penicillin/streptomycin, L-glutamine, poly-D-lysine, phosphate-buffered saline (PBS), DAPI were purchased from Sigma Aldrich (Milan, Italy) andPolyethyleneimine (MW 25,000 Da) from, Polysciences, Inc., (Tebu-Bio srl, Milan, Italy). Lipofectamine 2000 (Lipo) was from Invitrogen Life Technologies (Thermo Fisher Scientific, Grand Island, NY, USA). Enzymes used for molecular cloning and the enhanced chemiluminescence detection kit were from Roche Molecular Biochemicals (Merck KGaA, Darmstadt, Germany). All other chemicals used for SDS-poly-acrylamide gel electrophoresis and Western blotting were purchased from Bio-Rad laboratories (Hercules, CA, USA). Primary antibody used: mouse polyclonal antibody anti-PKR2 (sc-365696) from Santa Cruz Biotechnology (Aurogene srl, Rome, Italy) for Western blotting, rabbit polyclonal antibody against PKR2 (APR-042) from Alomone Labs (Aurogene srl, Rome, Italy) for immunofluorescence; mouse monoclonal antibody anti-FLAG (F1804) and anti-polyHistidine−Peroxidase (A7058) from Sigma Aldrich (Milan, Italy), rabbit polyclonal antibody anti-ERK (#44-654) and anti-pERK (#44-680) from Invitrogen (Thermo Fisher Scientific, Grand Island, NY, USA), secondary anti-species IgG antibodies coupled to Alexa Fluor-555 (Immunological Sciences, Rome, Italy). Nitrocellulose membranes (0.45 μm; Hybond-C Extra) were from Amersham Pharmacia Biotech (Milan, Italy). Chromatography resin types: His-tagged protein purification kit from Novagen (Merck KGaA, Darmstadt, Germany) and Glutathione-Sepharose beads and CM-Sephadex C-25 column from GE Healthcare (Aurogene srl, Rome, Italy), Anti-DYKDDDDK Affinity Resin from Sigma Aldrich (Milan, Italy). Cross-linker dithiobis(succinimidyl propionate) was from Sigma Aldrich (Milan, Italy).

### 2.2. Expression Constructs

For the bioluminescence resonance energy transfer (BRET) assay in mammalian culture cells, construction of green fluorescent protein (GFP)-β-arrestin-2, PKR2-rLuc, PKR1-rLuc plasmid were described in [14] and PKR2-ΔC52 rLuc was described in [38].

For crosslinking experiments to express the PKR2-ΔC52 mutant, we used PKR2-ΔC52 pcDNA described in [38] and to express MRAP2, we used the RC203668 plasmid purchased from Origene (Thermo Fisher Scientific, Grand Island, NY, USA).

For the expression of hMRAP2-Glutathione S-Transferase (GST), hMRAP2-CTpBS was digested with BamHI and EcoRI, and the obtained fragment was cloned into pGEX-2T. For production of the 131CT-MRAP2 deletion mutant in *E. coli*, we used pet 28-131CT-MRAP2 described in [39].

To express the 131CT-MRAP2 deletion mutant in a mammalian cell culture, the construct pet 28-131CT-MRAP2 was digested with EcoRI and HindII, and the obtained fragment was cloned into pCMV.

For the expression of β-arrestin-2 and p44 with His TAG in *E. coli*, we used a construct β-arrestin pET28 and p44 pET28 described in [38]. The β-arrestin-2 GST protein was obtained by cloning the fragment obtained by digestion of the β-arrestin-2 pET28 construct with BamHI and EcoRI in pGEX-2T.

### 2.3. Expression and Purification of Recombinant Proteins in E. coli

*E. coli* cultures expressing recombinant proteins were grown to optical density (O.D.) corresponding to 0.6. Cultures were grown for an additional 4 h at 28 °C in the presence of isopropyl-β-D-1-thiogalactopyranoside (IPTG) 0.1 mM. β-arrestin 2, p44, CT-MRAP2, and 131CT-MRAP2 were purified using nickel metal affinity chromatography (Ni-NTA His Bond resin). The purified recombinant proteins were separated by SDS-PAGE and analysed by Western blotting.

### 2.4. Glutathione S-Transferase (GST) Pull-Down

After IPTG-induced expression in *E. coli*, GST fusion proteins (β-arrestin-2-GST and hMRAP2-GST) were purified from cell extracts using affinity chromatography with glutathione-Sepharose beads according to the manufacturer's instructions. Briefly, 50 μL slurry of glutathione-Sepharose beads equilibrated in buffer A (PBS, 1% Nonidet P 40, 1 mM EDTA supplemented with protease inhibitor) was incubated with 2 mL CT-GST lysate for 60 min at 4 °C with constant stirring. Beads with bound GST proteins were collected by centrifugation, washed extensively in buffer A, and incubated with each purified protein β-arrestin-2, p44 or CT-MRAP2 and 131CT-MRAP2 expressed in *E. coli*. Beads were washed as described above, and bound proteins were eluted with GSH and analysed by Western blotting.

### 2.5. Cell Cultures, Transfection and Stimulation

CHO (Chinese hamster ovary) cells were plated at a density of $2 \times 10^5$ per well in 6-well plates or $4 \times 10^4$ per well in 24-well plates on poly-D-lysine coated coverslips and allowed to grow until they reached 70–80% confluence. Cells were grown in DMEM/F-12 containing 100 U/mL penicillin/streptomycin, 10% FBS and 2 mM L-glutamine at 37 °C and 5% $CO_2$. Transient transfection with plasmids was performed in the presence of Lipo according to the manufacturer's instructions. In brief, cells were incubated with the Lipo-DNA complex for 24 h at 37 °C and 5% $CO_2$. A total of 24 h after transfection, cells were deprived of serum and stimulated with PK2 (100 nM) for 10 min and 60 min at 37 °C and 5% $CO_2$. At the end of the incubation period, cells were lysed in RIPA lysis buffer containing a protease inhibitor cocktail (1% *v/v*), quantified, and used for Western blotanalysis or used for immunofluorescence analysis. Alternatively, transient transfections were carried out using linear polyethyleneimine as described in [40] and allowed to express the genes for 48 h before starting. CHO cell line stably expressing rGFP-fused-β-arrestin-2 chimera was obtained by the use of retroviral vectors (pQ serie Clontech, Takara Bio Saint-Germain-en-Laye, France), as indicated by the manufacturer.

### 2.6. Ligand Production

In *P. pastoris*, PK2 was expressed and purified, as described in Miele et al. [16]. Briefly, the induction of the synthesis was carried out for 120 h in BMMY medium daily supplemented with 1% methanol. The crude cultures were centrifuged to remove the cells. Supernatants were diluted 1:5 and applied to a CM-Sephadex C-25 column in 20 mM BES, pH 7.0. The elution of the ligands was performed with 20 mM BES, pH 7.0/0.2 M NaCl. Recombinant ligands were pooled, dialyzed against 20 mM Tris–HCl (pH 7.0) buffer and analysed in 18% (*w/v*) polyacrylamide SDS–PAGE gel.

### 2.7. Crosslinking

CHO cells were transiently transfected with β-Arrestin pCMV and PKR2 pcDNA or PKR2-ΔC52 pcDNA. After transfection (24 h), cells were incubated with vehicle or 100 nM PK2 for 15 min at 37 °C. Stimulations were stopped by adding the membrane-permeable and cross-linker dithiobis(succinimidyl propionate) at a final concentration of 2 mM. Cells were then incubated with gentle shaking at room temperature, washed twice with 50 mM Tris/HCl, pH 7.4, in PBS to neutralize unreacted dithiobis (succinimidyl propionate), and lysed in 0.5 mL 50 mM Hepes, pH 7.4, 50 mM NaCl, 10% (*v/v*) glycerol, 0.5% (*v/v*) Nonidet P40, 2 mM EDTA, 100 μM $Na_3VO_4$, 1 mM PMSF, 10 μg/mL benzamidine, 5 μg/mL soybean trypsin inhibitor, and 5 μg/mL leupeptin lysed, and clarified by centrifugation. Aliquots (25 μL) of whole-cell lysates were collected and mixed with an equal volume of 2× reducing loading buffer. Immunoprecipitates were washed 3 times with glycerol lysis buffer and eluted in 1× reducing loading buffer for 15 min at 45 °C.

### 2.8. Immunoprecipitation

Samples were immunoprecipitated using a FLAG bind resin following the batch purification under native conditions. Eluates were fractionated by SDS–PAGE on a 10%

gel and transferred to a polyvinylidene difluoride membrane. The membrane was probed with anti-FLAG and anti-His monoclonal antibody.

### 2.9. Western Blotting

After electrophoretic separation, the proteins were transferred onto nitrocellulose membrane (TCM) and were blocked in 1% non-fat Milk, 1% BSA/Tris-buffered saline with 0.10% Tween-20 (TBS-T pH 7.4) for 1 h at room temperature. Then, the membranes were incubated overnight at 4 °C with the appropriate primary antibody, in the blocking solution. After extensive washing with TBS-T, the membranes were incubated with anti-mouse or anti-goat IgG, HRP-linked secondary antibody for 1 h at room temperature. Immunoreactive signals were visualized with an enhanced chemiluminescence system.

### 2.10. Bioluminescence Resonance Energy Transfer (BRET) Assay

BRET assays monitoring receptor/β-arrestin-2 interaction were performed on CHO cells stably expressing rGFP-β-arrestin-2 after transient transfection with a constant amount of either receptors in addition to an increasing amount of MRAP2 encoding vectors. Cells were incubated with transfection mixture for 6 h before plating in 96-wells (OptiPlate, PerkinElmer, Rotterdam, The Netherlands). After 48 h incubation, a BRET assay was performed using a plate luminometer (VICTOR light, PerkinElmer, Rotterdam, The Netherlands).

Briefly, culture medium was replaced with 40 μL PBS containing 5 μM coelenterazine (CTZ), cells incubated for 10 min before the addition of 10 μL of a 5× concentrated ligand solution. Readings were taken after 5 min of further incubation. The BRET ratio was determined as the ratio of high-energy (donor) and low-energy (acceptor) emissions recorded sequentially with different filters, as described [14]. Concentration–response curves were analysed by nonlinear curve fitting to the general logistic function: $y = (a - d)/[1 + (x/c)b] + d$, where y and x are the BRET ratio and ligand concentration, a and d are the upper and lower asymptotes, c is the ligand concentration giving the half-maximal BRET change, and b is the slope factor at c [14,38].

### 2.11. Immunofluorescence

After the stimulation, CHO cells were washed in PBS 1×, blocked for 1 h in 5% normal donkey serum and then incubated overnight at 4 °C with a rabbit polyclonal antibody against PKR2. Cells were then washed with PBS 1× and incubated for 1 h at room temperature with secondary anti-species IgG antibodies coupled to Alexa Fluor-555 (Immunological Sciences, Rome, Italy). The cell nuclei were stained with DAPI. The fluorescence signal was recorded using an Eclipse E600 fluorescence microscope (Nikon Instruments, Tokyo, Japan) connected to a high-resolution digital camera (Nikon Digital Sight DSU1) with 64-bit NIS-Elements BR 3.2 software.

### 2.12. Data Analysis

The data were plotted and analysed using GraphPad Prism 7 for Windows. All results were expressed as the mean ± SEM. Statistical analyses were performed using two-way ANOVA followed by Sidak's multiple comparisons test, one-way ANOVA followed by Dunnett's multiple comparisons post-test, and Students' *t*-test. Differences were considered significant at $p < 0.05$.

## 3. Results

### 3.1. MRAP2 Inhibits PK2-Stimulated β-Arrestin-2 Recruitment to PKRs

CHO cells were transfected with plasmids encoding PKR2 and β-arrestin-2 fused with GFP-β-arrestin-2 in the presence or absence of a plasmid encoding MRAP2. The cells were stained after 10 min of treatment with vehicle or 100 nM PK2. It is known that β-arrestin-2 and PKR2 colocalise in the endoplasmic reticulum in the absence of a ligand and treatment with PK2 induces the translocation of PKR2 and GFP-β-arrestin-2 to the plasma membrane [38,41]. In the absence of PK2, the co-expression of MRAP2

induces the localisation of PKR2 and GFP-β-arrestin-2 to the plasma membrane, which is not increased by the addition of PK2 (Figure 1A). The lack of β-arrestin-2 recruitment after PKR2 activation in the presence of MRAP2 was confirmed by BRET assays between luminescent PKR2 (PKR2-rLuc) and fluorescent β-arrestin-2 (GFP-β-arrestin-2). Using these assays, we also found that in the absence of PK2 ligand, the value of the BRET ratio, which relates to the constitutive association between PKR2 and β-arrestin-2, increases with increasing MRAP2 concentration (Figure 1B). The expression of β-arrestin-2-GFP is almost constant and is not affected by the presence of MRAP2 (Figure 1C). On the contrary, the expression of PKR2 is modulated by the presence of MRAP2. In particular, the reduction of about 45% in the presence of a receptor/MRAP2 ratio of 1:15 confirms the data already available in the literature [29] (Figure 1D). However, our data emphasise that in the presence of a receptor/MRAP2 ratio of 1:5, although there is no strong reduction in receptor expression (approximately 20%), we observe a drastic decrease in β-arrestin-2 recruitment in the presence of PK2 ligand and a concomitant increase in β-arrestin-2-PKR2 interaction in the absence of PK2 ligand (basal binding).

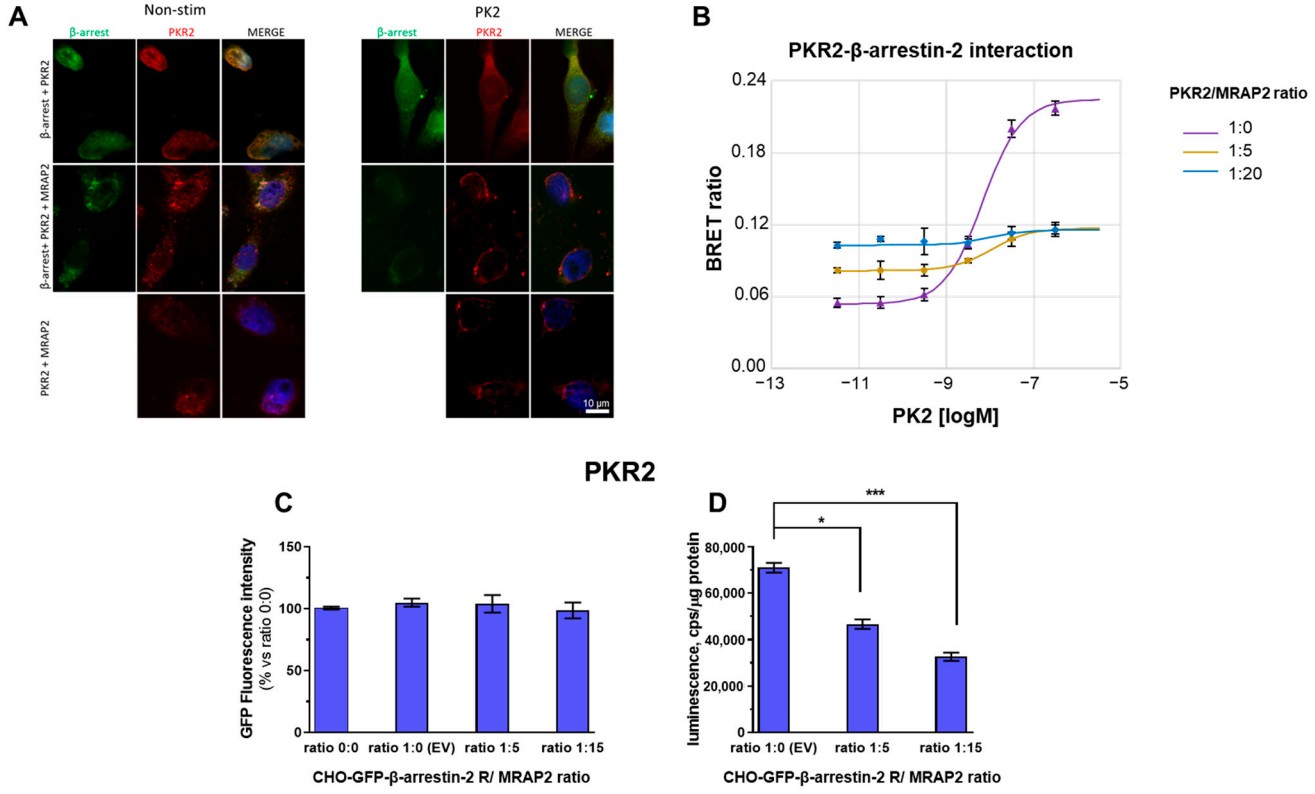

**Figure 1.** β-arrestin-2 recruitment to PKR2 in the presence of MRAP2. (**A**) Representative immunofluorescence images of CHO cells transfected with GFP-β-arrestin-2 (green) and PKR2 (red), in the absence or in the presence of MRAP2 and in the absence of PK2 (−) or in the presence of PK2 100 nM (+) for 1 h. Scale bar 10 μm. The cell nuclei were counterstained with DAPI (blue). (**B**) Concentration–response curves of PK2 after transient transfection of PKR2-rLuc recorded by BRET assay in CHO cells stably co-expressing GFP-β-arrestin-2 and transiently transfected with PKR2-rLuc and with increasing amounts of MRAP2 vectors (PKR2-rLuc/MRAP2 ratio: 1:0, 1:5, or 1:20). (**C**) GFP β-arrestin-2 levels were analysed on cell monolayers of transfected cells expressing rGFP-β-arrestin-2 after transient transfection with equal amounts of the luminescent receptor and increasing amounts of vectors expressing MRAP2. (**D**) The levels of luminescent receptor PKR2/rLuc was measured on total protein preparations (cps/μg proteins) of cells expressing rGFP-β-arrestin-2 after transient transfection with equal amounts of either luminescent receptor or increasing amounts of vectors expressing MRAP2 using a plate luminometer. The bar graph represents the mean ± SEM of the triplicate determination

of cps values obtained in each individual experimental condition (receptor/MRAP2 cDNA ratio: 1:0; 1:5; 1:15). Data (symbols) representing the means $\pm$ SEM of three independent determinations in the contest of one over two independent experiments performed, giving overlapping results (means of triplicates), are shown with the best-fitting theoretical curves. A student's t-test was used for statistical analysis; * $p < 0.05$, *** $p < 0.001$.

To determine whether the negative effect of MRAP2 on the recruitment of β-arrestin-2 to PKR2 is not simply due to the inhibition of MRAP2-induced PKR2 translocation to the plasma membrane, BRET assays were performed between luminescent PKR1 (PKR1-rLuc) and fluorescent β-arrestin-2 (GFP-β-arrestin-2). It is known that MRAP2 inhibits the localisation of PKR1 to the membrane significantly less than that of PKR2 [29]. BRET experiments show that MRAP2 is able to inhibit the association of β-arrestin-2 with PKR1 at the plasma membrane with an effect very similar to that described for PKR2 (Figure 2A). Finally, to confirm the result, the recruitment of β-arrestin-2 was examined in the presence of a MRAP2 mutant obtained by deleting the C-terminal region from residue 131 to the end of the MRAP2 protein. The 131CT-MRAP2 mutant does not retain the ability to modulate PKR1 activity but is able to alter the localisation of the receptor [42]. The results of the BRET assays performed between luminescent PKR1 (PKR1-rLuc) and fluorescent β-arrestin-2 (GFP-β-arrestin-2) in the presence of the 131CT-MRAP2 mutant show that the mutant does not inhibit the recruitment of β-arrestin-2 to the membrane (Figure 2B), demonstrating that MRAP2 inhibits β-arrestin-2 recruitment and does not only alter PKR1 localisation to the membrane.

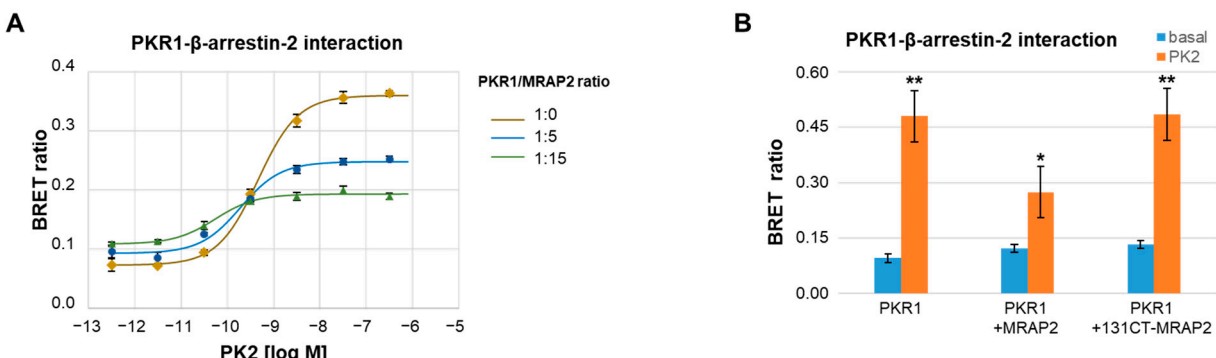

**Figure 2.** Role of MRAP2 on β-arrestin-2 recruitment to PKR1. (**A**) Concentration–response curves of PKR1-rLuc recorded by BRET assay in CHO cells stably co-expressing GFP-β-arrestin-2 in the absence or presence of MRAP2. Concentration-dependent increase of PKR1-β-arrestin-2 interaction measured in CHO cells stably expressing GFP-β-arrestin-2 after transient transfection with constant amount of PKR1-rLuc in association with either empty or increasing amounts of MRAP2 at a 1:5 or 1:15 receptor/MRAP2 cDNA ratio. (**B**) Bar graphs of PKR1/rLuc recorded by BRET assay in CHO cells stably expressing GFP-β-arrestin-2 transfected with a constant amount of PKR1-rLuc plus either empty or MRAP2 or 131CT-MRAP2 mutant expressing vectors (1:15 ratio) in the absence of PK2 (−) or after 10 min incubation with PK2 100 nM. Data are the means $\pm$ SEM of triplicate values determined in one representative over three independent experiments performed. A Students' *t*-test was used for statistical analysis; * $p < 0.05$, ** $p < 0.01$ versus basal.

### 3.2. Role of PKR2 C-Terminal Region in MRAP2 Modulation of β-Arrestin-2 Recruitment

To determine the role of the C-terminal region of PKR2 in MRAP2 modulation of β-arrestin-2 recruitment, we used a PKR2 mutant obtained by deleting 52 amino acids in the C-terminal region of PKR2 (PKR2-ΔC52 mutant). To demonstrate the ability of MRAP2 to physically interact with the PKR2-ΔC52 mutant, we first expressed WT PKR2 and the PKR2-ΔC52 mutant together with MRAP2 in CHO cells. The cells were incubated with the cross-linker, and the membrane proteins were fractionated by SDS-PAGE, blotted, and then analysed with the anti-PKR2 antibody. The result shown in Figure 3A evidences the clear formation of a complex of MRAP2 with WT PKR2 or PKR2-ΔC52. The molecular

weight of the complex is ~70 kDa, being the un-combined molecular weight of MRAP2 and PKR2 ~25 kDa and 45 kDa, respectively. The interaction of β-arrestin-2 with the PKR2-ΔC52 mutant was then analysed using the BRET assay. The results show that MRAP2 increases the binding of β-arrestin-2 to the PKR2-ΔC52 mutant in a concentration-dependent manner and with a more pronounced effect compared to WT PKR2 (Figure 3B). The GFP β-arrestin-2 levels were comparable in cells expressing rGFP-β-arrestin-2 after transient transfection with equal amounts of the luminescent receptor and increasing amounts of vectors expressing MRAP2 (Figure 3C).

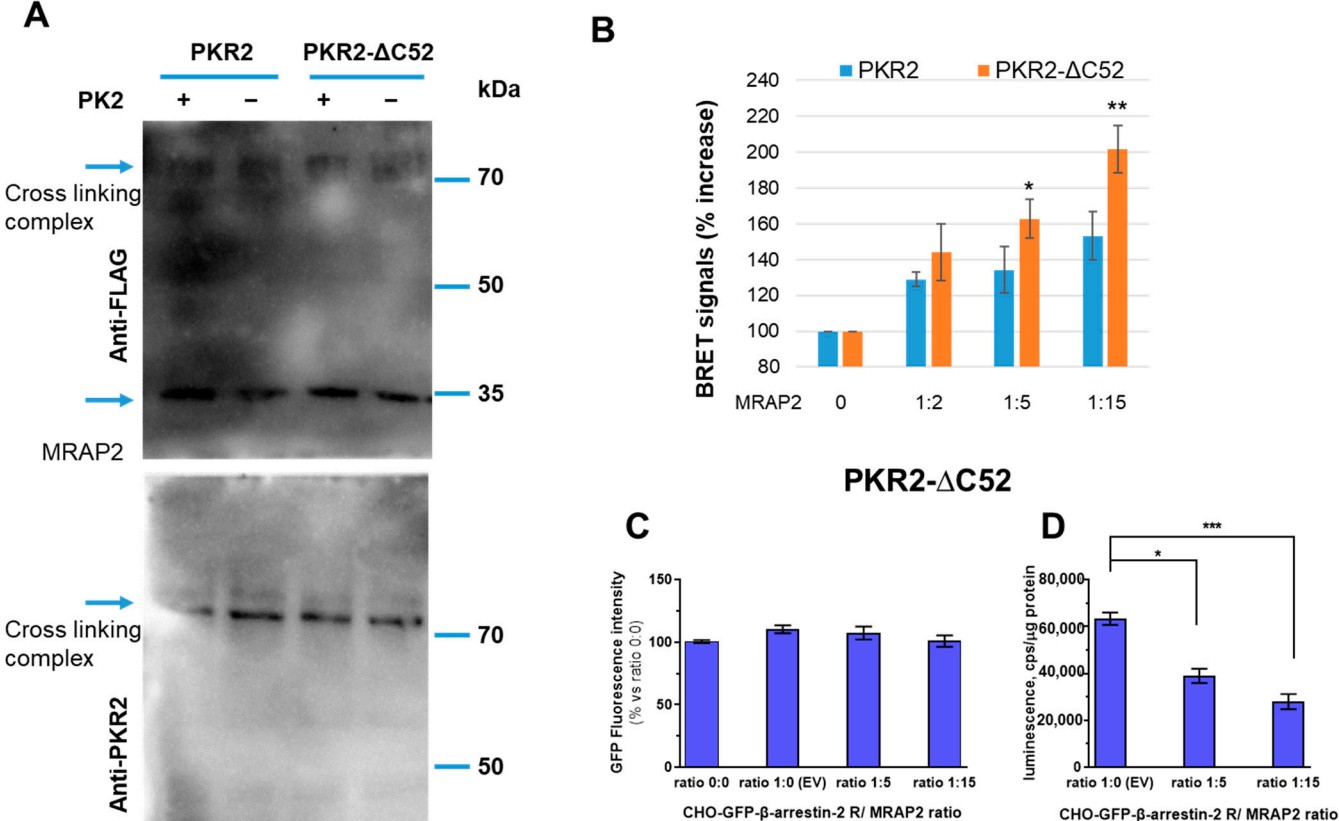

**Figure 3.** The C-terminal region of PKR2 in the modulation of β-arrestin-2 recruitment by MRAP2. (**A**) Cross-linking of MRAP2 with PKR2 or the PKR2-ΔC52 receptor mutant. CHO cells expressing WT PKR2 or the PKR2-ΔC52 mutant were incubated with dithiobis (succinimidyl propionate). Proteins were immunoblotted and probed with anti-PKR2 antibodies. (**B**) Bar graphs of PKR2-rLuc and PKR2-ΔC52 mutant-rLuc recorded by BRET assay in CHO cells stably co-expressing GFP-β-arrestin-2 in the absence or presence of MRAP2. PKR2 and PKR2-ΔC52-βarrestin-2 interaction in the presence of increasing concentrations of MRAP2 (1:2, 1:5, and 1:15 receptor/MRAP2 ratio), measured in the absence of PK2. A Students' *t*-test was used for statistical analysis; * *p* < 0.05, ** *p* < 0.01 versus basal. (**C**) GFP β-arrestin-2 levels were analysed on cell monolayers of transfected cells expressing rGFP-β-arrestin-2 after transient transfection with equal amounts of the luminescent PKR2-ΔC52 receptor and increasing amounts of vectors expressing MRAP2. (**D**) The levels of luminescent receptor PKR2-ΔC52/rLuc were measured on total protein preparations (cps/μg proteins) of cells expressing rGFP-β-arrestin-2 after transient transfection with equal amounts of either luminescent receptor or increasing amounts of vectors expressing MRAP2 using a plate luminometer. The bar graph represents the mean± SEM of the triplicate determination of cps values obtained in each individual experimental condition (receptor/MRAP2 cDNA ratio: 1:0; 1:5; 1:15). Data are expressed as the percentage of the signal increase in the presence of MRAP2 compared to the empty vector. Data are the means ± SEM from three independent experiments, each performed in triplicate. One-way ANOVA was used for statistical analysis followed by a Dunnett's test for multiple comparisons * *p* < 0.05, *** *p* < 0.001 vs. PKR2-ΔC52 MRAP 0.

MRAP2 modulates the expression of the PKR2-ΔC52 mutant similarly to that of PKR2, so it is possible to compare the effects of MRAP2 on β-arrestin-2 binding to the two receptors (Figure 3D).

### 3.3. MRAP2 Inhibits β-Arrestin–Dependent Erk Activation

To provide further evidence for the modulation of β-arrestin-2 binding to WT PKR2 by MRAP2, ERK activation was examined in the presence of MRAP2. By transfecting MRAP2 into CHO cells expressing WT PKR2, no increase in phospho-ERK1/2 levels was observed at either 10 or 60 min. The results showed that in the presence of MRAP2, PKR2-induced ERK activation is blocked, suggesting that MRAP2 inhibits ERK activation mediated by the G protein (evaluable at 10 min) and by β-arrestin-2 (evaluable at 60 min) (Figure 4) [38].

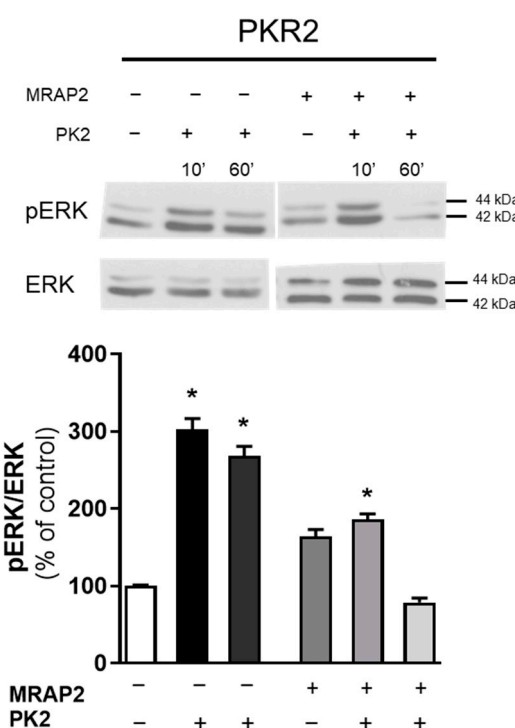

**Figure 4.** Analysis of $ERK_{1/2}$ phosphorylation in CHO cells transfected with WT PKR2. The densitometric plots show the $pERK_{1/2}$ and $ERK_{1/2}$ protein levels at 10 and 60 min after treatment with PK2 100 nM. The bar graphs show the ratio of $pERK_{1/2}/ERK_{1/2}$ and the percentage increase compared to unstimulated cells (control). The bars show the mean values ± SEM from the three experimental conditions. A two-way ANOVA followed by Sidak's multiple comparisons test was used for statistical analysis * $p < 0.05$ versus control.

### 3.4. MRAP2 Physical Interact with β-Arrestin-2

To determine whether the modulation of β-arrestin-2 recruitment to PKR2 is due to a direct interaction between β-arrestin-2 and MRAP2, we analysed the physical interaction between MRAP2 and β-arrestin-2 using two different Glutathione S-Transferase (GST) pull-down experiments. In the first experiment, the β-arrestin-2-GST fusion protein was purified on glutathione agarose and then used in direct binding experiments with the two MRAP2 deletion mutants CT-MRAP2 and 131CT-MRAP2 with His-tag (Figure 5A). CT-MRAP2 contains a region extending from the residue at position 78 to the residue at position 204 and the 131CT-MRAP2 mutant contains only a part of the C-terminal region between the residue at position 78 and the residue at position 131. The results show that CT-MRAP2 and 131CT-MRAP2 are able to bind β-arrestin-2. As already reported CT-MRAP2 and 131CT-MRAP2 are not only bound to GST [39]. Similarly, in the second experiment, the C-terminal domain of MRAP2 (the region from the residue at position 78 to the residue at position 204) was fused to GST (hMRAP2-GST), purified on glutathione agarose, and then

used in direct binding experiments of β-arrestin-2 or p44 labelled with a His-tag (Figure 5B). p44 is the splicing variant of β-arrestin-2, with deletion of C terminal tail, with a predicted molecular weight of ~44 kDa able to assume a constitutively active conformation [43]. The results show that β-arrestin-2 and p44 are able to bind the C-terminal domain of MRAP2.

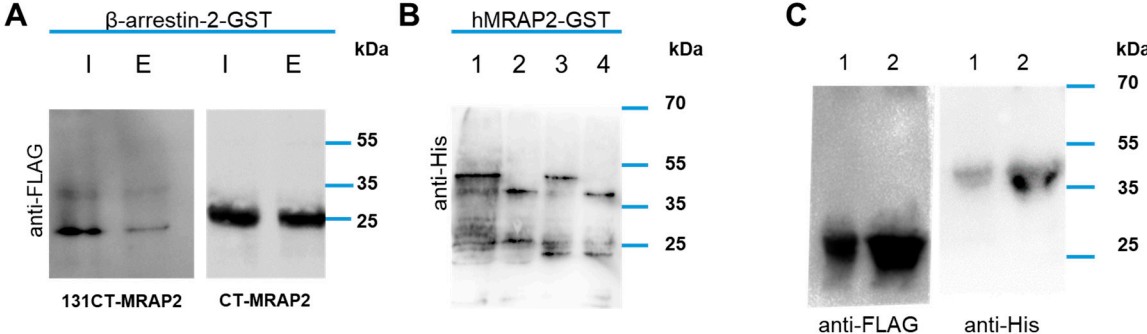

**Figure 5.** MRAP2 interaction with β-arrestin-2. (**A**) The GST-fusion protein β-arrestin-2-GST was used to pull down CT-MRAP2 or 131CT-MRAP2 FLAG-tagged. The resulting elution solutions were resolved on 12% SDS-PAGE and analysed by Western blotting with an anti-anti FLAG antibody. I, input; E, eluate. (**B**) GST-fusion protein hMRAP2-GST was used to pull down β-arrestin-2 or p44. The resulting elution solutions were resolved on 12% SDS-PAGE and analysed by Western blotting with an anti-His antibody. 1. β-arrestin-2 input; 2. p44 input; 3. β-arrestin-2 eluate; 4. p44 eluate. (**C**) Co-immunoprecipitation experiments using β-arrestin-2 or p44 with His-Tag and CT-MRAP2 with FLAG-tag. Total proteins from *E. coli* co-expressing CT-MRAP2 FLAG tagged and either β-arrestin-2 (line 1) or p44 (line 2) His-tagged were immunoprecipitated using a FLAG binding resin and the immunoprecipitates resolved by SDS-PAGE. The immunoblots were probed with anti-FLAG (left panel) and anti-His (right panel) antibodies.

To confirm this result, we performed immunoprecipitation experiments by the expression of His-tag labelled β-arrestin-2 or p44 and the FLAG-tag labelled C-terminal domain of MRAP2 (CT-MRAP2 FLAG). CT-MRAP2 was immunoprecipitated, subjected to SDS–PAGE, and immunodetected using a commercial polyclonal antibody raised against a His tag. Interestingly, β-arrestin-2 and p44 are able to co-immunoprecipitate with the C-terminal domain of MRAP2, as highlighted (Figure 5C).

## 4. Discussion

G protein-coupled receptors (GPCRs) represent the largest superfamily of cell surface receptors capable of transmitting a variety of extracellular signals in cells. The binding of ligands leads to a conformational change of the receptor, which results in the binding of intracellular signalling transducers such as G-proteins or β-arrestins to the intracellular part of the receptor [44–46]. The recruitment and stability of GPCR-β-arrestin interactions depend on the affinity of the β-arrestin for two GPCR binding sites. The first binding, the so-called tail interaction, occurs between the cytosolic tail of a GPCR and the N-terminal domain of β-arrestin [44] and allows β-arrestin to reach its active conformation. Conversely, p44, a splice variant of β-arrestin, adopts the active conformation constitutively before the binding of the GPCR tail [43]. The second interaction that occurs between the central β-arrestin ridge and the cytoplasmic loops, mainly intracellular loops 2 and 3 of the receptor, is often referred to as core interaction [44]. It is now clear that receptor activation is a highly dynamic process, as the receptor can adopt different active conformations that can trigger different signalling pathways. The stability of the different conformations determines the preferential activation of one signalling pathway over another. The balance between the conformations can be shifted in favour of one conformation by the binding of different ligands, by the presence of mutations of the receptor, or by the binding of accessory proteins [45].

MRAP2 is a single transmembrane protein expressed on the cell surface and reticulum membrane mainly in the stomach and endocrine glands, hypothalamus, and adipocytes [21]. MRAP2 modulates various GPCRs that are important for energy homeostasis, such as the melanocortin-4 receptor, orexin, and ghrelin. MRAP2 also binds and regulates prokineticin receptors (PKRs) by reducing their signalling pathway and plasmatic membrane localisation [29,39,42,47].

Our data show that MRAP2 inhibits the recruitment of β-arrestin-2 to PKRs in the presence of ligands and ERK activation via the β-arrestin-dependent pathway.

In contrast, it does not inhibit basal binding, which occurs in the absence of ligands. Immunofluorescence experiments show that MRAP2 enhances the colocalisation of β-arrestin-2 with PKR2 on the membrane in the absence of ligand, but after stimulation with the PK2 ligand, MRAP2 prevents the localisation of PKR2 and β-arrestin-2 on the membrane. Since it is known from the literature that MRAP2 inhibits the expression of PKR2 and its localisation on the membrane [29], we wanted to exclude the possibility that the absence of β-arrestin on the membrane after treatment with the ligand was due to the lack of the localisation of PKR2 on the membrane by MRAP2. Therefore, we performed the BRET experiments with prokineticin receptor 1 (PKR1), whose localisation is only weakly inhibited by the presence of MRAP2 [29]. The results show that MRAP2 cannot enhance the basal binding of β-arrestin-2 to PKR1 but is able to inhibit the association of β-arrestin-2 with PKR1 induced by PK2. We then repeated the BRET assay with PKR1 in the presence of the MRAP2 mutant obtained by deletion of the C-terminal region, termed 131CT-MRAP2. The 131CT-MRAP2 mutant is unable to affect the recruitment of β-arrestin-2 to PKR1, but inhibits receptor trafficking [38,41] without altering the activity of PKR1 [39,42,47].

We therefore analysed the modulation of the basal binding of β-arrestin-2 to the PKR2 receptor in the presence of MRAP2. Using the PKR2-ΔC52 mutant obtained by deletion of the C-terminal region of PKR2, we have previously shown that the basal interaction of β-arrestin-2 with PKR2 is not mediated by the C-terminal region of PKR2 [38]. In this study, we first used cross-linking experiments to show that the mutant can bind MRAP2. Using BRET assays, we were therefore able to show that in the presence of MRAP2, the basal binding between β-arrestin-2 and the mutant receptor PKR2-ΔC52 is enhanced. Furthermore, the effect of MRAP2 on the PKR2-ΔC52 mutant is stronger than on the WT-PKR2 receptor and increases with increasing concentration of MRAP2. The data obtained suggest that the presence of the C-terminus prevents MRAP2 from forming the basal interaction of the receptor with β-arrestin-2. Therefore, this work highlights the role of MRAP2 in modulating PKR2 β-arrestin-2 recruitment and β-arrestin-2-mediated signalling, as previously shown for ghrelin receptors [48–50]. Furthermore, GST pull-down and immunoprecipitation show that β-arrestin-2 and p44, the active form of β-arrestin-2, bind directly to a specific region of MRAP2. Our data show that the 131CT-MRAP2 mutant, which contains the region from residue 1 to residue 131 of MRAP2, is able to interact with β-arrestin-2, although it is unable to prevent the recruitment of β-arrestin-2 to PKRs.

## 5. Conclusions

This work suggests that MRAP2 can be regarded as an allosteric regulator of PKR2. This type of regulator binds an allosteric site that is distinct from the orthosteric site recognised by endogenous ligands and stabilises a specific receptor conformation. The characterisation of the natural allosteric modulation of GPCRs is a promising new approach for GPCR drug discovery compared to classical approaches that focus on the development of small molecules targeting the orthosteric site [51–56].

**Author Contributions:** R.L. and R.M. were responsible for the overall conception of the project, planned the experiments and supervised, edited and contributed to the critical revision of the manuscript. I.C., M.R.F., D.M. and M.V. conducted the experiments and analysed the data. All authors have read and agreed to the published version of the manuscript.

**Funding:** This work was supported by intramural grants "Avvio alla ricerca 2022—tipo 1" (N°AR12218162ED57D1) and "Avvio alla ricerca 2023—tipo 2" (N°AR223188B489287F) from Sapienza University of Rome to M.V.; and by Fondazione Sovena, Rome, Italy (fellowship to M.V.).

**Institutional Review Board Statement:** Not applicable.

**Informed Consent Statement:** Not applicable.

**Data Availability Statement:** The data presented in this study are available on request from the corresponding author.

**Conflicts of Interest:** The authors declare no conflicts of interest.

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
