# Peer review of "MRAP2 Inhibits β-Arrestin-2 Recruitment to the Prokineticin Receptor 2"

_cimb, doi:10.3390/cimb46020104_

Round 1
Reviewer 1 Report
Comments and Suggestions for Authors
The manuscript „MRAP2 inhibits β-arrestin-2 recruitment to the Prokineticin Receptor 2” describes a specific set of interactions between prokineticin receptors, prokineticins, β-arrestins and Melanocortin receptor accessory protein 2 (MRAP2).
1. The text is concise and full of information. It is not easy to follow all the interactions if one is not a specialist in this field. After reading the introduction an idea of drawing a scheme of interactions was really tempting. It does not help that the Authors use abbreviations for most of the analyzed ligands and receptors, distinguishing MRAP and MAPK, PK1, PKR1 etc. requires a lot of attention.
2. In some cases the description may cause misunderstandings: “It is known that β-arrestin-2 and PKR2 colocalise in the endoplasmic reticulum in the absence of ligand and treatment with PK2 induces translocation of PKR2 and GFP-β-arrestin-2 to the plasma membrane [27,30].” The GFP version may seem as a different form affected by PK2, please consider providing explanation for the use of GFP modification.
3. Some abbreviations are explained latter in the text, or, in the case of experimental part, no explanation is offered (BRET, GFP, O.D., CTZ, …)
4. Two sentences at the end of introduction summarize the results, after the aim of work is presented. Please reconsider, a plan of the study could be more beneficial. (We demonstrated that MRAP2 inhibits ß-arrestin -2 recruitment and ß-arrestin-mediated signalling. The data suggest that MRAP2 physically interacts with ß-arrestin and stabilises the basal interaction of ß-arrestin-2 with PKR2.)
5. The discussed molecules are proteins, their size (kDa) could be helpful in following the results (gel position, especially after cross-linking).
6. Please verify the time: stimulated with PK2 (100 nM) for 10 minutes and 1 hour at 37°C and 5% CO2.
7. Please specify the controls used (vehicle, vector etc. in experimental descriptions).
8. Please reconsider the scale of Y axis in Fig. 3B (is control a 100% increase?)
9. In general, it is difficult to follow the results without a careful study of the experiments, to understand their idea and reason of use. Some parts of the results are complicated by a strange structure of sentences (…we analysed by two different experiments the physical interaction between MRAP2 and β-arrestin-2 using GST pull-down. It was performed two diverse experiment , In the first experiment, The GST-β-arrestin-2 fusion protein was purified on glutathione agarose…)
10. Protein sequence is not encoded by amino acid residues, please rewrite the sentence (C-terminal region encoded by the residue at position 78 to the residue at position 131)
11. The discussion is much easier to read, still, the reason for some experiments could be explained in more detail.
12. The conclusion: “a promising new approach for the development of therapeutics for the treatment of various diseases, offering advantages over orthosteric ligands” is quite general. Do the Authors consider possible protein ligands, mimetics, small molecules or genetic interventions as the possible therapeutic forms?
Other comments:
In the description of experiments, only few reagents are mentioned in 2.1, other are added in specific descriptions. The source information should be presented in the same way in the whole text, as well as references to the procedures.
In Fig. 1 and 2, the legends for graphs (panels 1B and 2A) affect the idea of presentation as the lines look like parts of the graph (1B).
Language and editorial problems:
Page 2: Recently has been demonstrated that PK2,
In vivo (italics?), E. Coli or E. coli?
A spellcheck would be beneficial: SDS-poly- acrylamide gel electrophoresis; “The β-arrestin-2-GST protein was obtained by fusing cDNA coding for β-arrestin-2 with GST we use β-arrestin-2 pET28 digested”; “and incubated with each purified protein His tagged respectively ß-arrestin-2, p44 or CT-MRAP2 and 131CT-MRAP2 expressed in E. Coli.”; over- night
Please use one form of His-tag name and capital/not-capital letter in arrestin, and one version of ß-arrestin-2-GFP (Fig. 1 caption)
CO2.
Fragment of Fig. 5 caption (antibody.1. Input ß-arrestin-2 imput; 2. p44 input; 3. ß-arrestin…)
Comments on the Quality of English LanguageComments included in the partfor the Authors.
There are no actual language problems but rather some editorial errors.
Author Response
Reviewer 1
The manuscript, MRAP2 inhibits β-arrestin-2 recruitment to the Prokineticin Receptor 2” describes a specific set of interactions between prokineticin receptors, prokineticins, β-arrestins and Melanocortin receptor accessory protein 2 (MRAP2).
- The text is concise and full of information. It is not easy to follow all the interactions if one is not a specialist in this field. After reading the introduction an idea of drawing a scheme of interactions was really tempting. It does not help that the Authors use abbreviations for most of the analyzed ligands and receptors, distinguishing MRAP and MAPK, PK1, PKR1 etc. requires a lot of attention.1
Thank you for the suggestion. We have added a scheme of the interactions in the introduction. We checked the abbreviations used and they result explained in the text.
- In some cases the description may cause misunderstandings: “It is known that β-arrestin-2 and PKR2 colocalise in the endoplasmic reticulum in the absence of ligand and treatment with PK2 induces translocation of PKR2 and GFP-β-arrestin-2 to the plasma membrane [27,30].” The GFP version may seem as a different form affected by PK2, please consider providing explanation for the use of GFP modification.
Thank you. It is a well validated method the use of β-arrestin-2 fused to GFP to evaluate the β-arrestin-2 localization (Ferguson SG , Caron MG. Green fluorescent protein-tagged beta-arrestin translocation as a measure of G protein-coupled receptor activation S G Methods Mol Biol. 2004:237:121-6.doi: 10.1385/1-59259-430-1:121).
- Some abbreviations are explained latter in the text, or, in the case of experimental part, no explanation is offered (BRET, GFP, O.D., CTZ, …)
Thank you. We introduced the meaning of abbreviations.
- Two sentences at the end of introduction summarize the results, after the aim of work is presented. Please reconsider, a plan of the study could be more beneficial. (We demonstrated that MRAP2 inhibits ß-arrestin -2 recruitment and ß-arrestin-mediated signalling. The data suggest that MRAP2 physically interacts with ß-arrestin and stabilises the basal interaction of ß-arrestin-2 with PKR2.)
Thank you for your suggestion We modified the introduction according to your observations.
- The discussed molecules are proteins, their size (kDa) could be helpful in following the results (gel position, especially after cross-linking).
Thank you for your suggestion. We specified the sizes of the proteins in the results.
- Please verify the time: stimulated with PK2 (100 nM) for 10 minutes and 1 hour at 37°C and 5% CO2
In a previous paper (Lattanzi et al., International Journal of Molecular Sciences , submitted) we performed a PK2 time-course of ERK activation establishing that ERK activation induced by G-protein occurs at 10 min and ERK activation induced by b-arrestin-2 mediated occurs at 60 min.
- Please specify the controls used (vehicle, vector etc. in experimental descriptions).
We specified that controls are unstimulated cells in the western blot experiment for evaluation of ERK activation.
- Please reconsider the scale of Y axis in Fig. 3B (is control a 100% increase?)
Thank you. We modified the graph changing the scale of Y axis to evidence that the control is 100%.
- In general, it is difficult to follow the results without a careful study of the experiments, to understand their idea and reason of use. Some parts of the results are complicated by a strange structure of sentences (…we analysed by two different experiments the physical interaction between MRAP2 and β-arrestin-2 using GST pull-down. It was performed two diverse experiment , In the first experiment, The GST-β-arrestin-2 fusion protein was purified on glutathione agarose…)
Thank you. We modified some sentences to clarify the manuscript.
- Protein sequence is not encoded by amino acid residues, please rewrite the sentence (C-terminal region encoded by the residue at position 78 to the residue at position 131)
Thank you. We changed the phrase.
- The discussion is much easier to read, still, the reason for some experiments could be explained in more detail.
Thank you. We changed the discussion in the part regarding the comments of the results.
- The conclusion: “a promising new approach for the development of therapeutics for the treatment of various diseases, offering advantages over orthosteric ligands” is quite general. Do the Authors consider possible protein ligands, mimetics, small molecules or genetic interventions as the possible therapeutic forms?
Thank you. We changed the conclusion to render them clearer.
Other comments:
In the description of experiments, only few reagents are mentioned in 2.1, other are added in specific descriptions. The source information should be presented in the same way in the whole text, as well as references to the procedures.
Thank you. We mentioned other reagents in paragraph 2.1: drugs and reagents
In Fig. 1 and 2, the legends for graphs (panels 1B and 2A) affect the idea of presentation as the lines look like parts of the graph (1B).
Thank you. We moved the legends for graphs (panels 1B and 2A) outside the graphs.
Language and editorial problems:
Page 2: Recently has been demonstrated that PK2,
In vivo (italics?), E. Coli or E. coli?
A spellcheck would be beneficial: SDS-poly- acrylamide gel electrophoresis; “The β-arrestin-2-GST protein was obtained by fusing cDNA coding for β-arrestin-2 with GST we use β-arrestin-2 pET28 digested”; “and incubated with each purified protein His tagged respectively ß-arrestin-2, p44 or CT-MRAP2 and 131CT-MRAP2 expressed in E. Coli.”; over- night
Please use one form of His-tag name and capital/not-capital letter in arrestin, and one version of ß-arrestin-2-GFP (Fig. 1 caption)
CO2.
Fragment of Fig. 5 caption (antibody.1. Input ß-arrestin-2 imput; 2. p44 input; 3. ß-arrestin…)
Thank you. We corrected all language and editorial problems.

Reviewer 2 Report
Comments and Suggestions for Authors
An expanding role for melanocortin accessory proteins (MRAPs) is becoming increasingly evident. The function of these single-membrane-spanning proteins goes beyond the melanocortin system, and the interaction with an increasing list of GPCRs transcends the modulation of constitutive activity or receptor membrane chaperoning. For example, an interplay between arrestin recruitment and MRAP2 (the isoform predominant in the brain and other peripheral tissues outside the suprarenal glands) was described for ghrelin receptors and is the subject of the study presented by Lattanzi and collaborators herein. This manuscript argues that MRAP2 inhibits beta-arrestin-2 (a.k.a. arrestin-3) recruitment by the prokineticin receptor 2. This is a provocative proposition that posits a broader role for MRAPs in general as critical modulators of GPCR function.
To support their results, the authors perform cell-based IHC and BRET-based arrestin recruitment assays in the presence of increasing amounts of transfected MRAP2, pull-down experiments, and ERK phosphorylation as a surrogate of G-protein and arrestin-mediated signaling. At face value, the results presented by the authors seem to support their claim, but the lack of fundamental controls holds back these assertions.
Major concerns
1 .- Protein expression squelching is an artifact perniciously ignored in the literature, significantly leading to erroneous result misinterpretations. The effects of over-expression that ultimately affect overall protein expression should be considered when analyzing data. For example, increasing the concentration of MRAP2 by twenty-fold will have a non-linear (or easy to predict) impact on PKR2 and arrestin cell expression. A recognized limitation of BRET-based experiments is finding the most idoneous way to account for this effect. For example, if available, sometimes a negative or loss-of-function mutant of the protein being studied should be used in lieu of the use of no protein at all, as the relative expression of the other components of the system WILL be affected by the absence or presence of the protein being studied. In this regard, the authors do not show how the presence or absence of MRAP2 affects the expression of PKR2 or Arr. BRET assays are particularly sensitive to this artifact and WILL lead to erroneous data interpretation if not adequately accounted for. Since the truncated form of the PKR2 seems to indicate that arrestin recruitment is driven by the intracellular loops rather than the receptor C-terminal domain, it is unlikely that the truncated receptor could be used as proper control. To account for these artifacts, the authors could resort to careful equalization of fluorescence and luminescence absolute (or raw) readouts (which are directly proportional to protein expression), but most importantly, by demonstrating or quantifying the cell surface expression of PKRs and MRAPs (since this would not be feasible for arresting due to their cytosolic nature) by cell surface ELISAs, flow cytometry, or the appropriate method of their choosing.
2 .- Low-quality blots for pull-down and ERK phosphorylation experiments do not support the author's conclusions and (respectfully) seem detracted from the quantifications. The authors should present better-quality blots in a revised manuscript.
3 .- Inadequate statistical analyses. It stands to remind the authors that the methodology used to test an experimental hypothesis is determined a priori. Suppose an experiment was designed to test the effect of two continuous independent variables on a continuous dependent variable (e.g., ERK phosphorylation). In that case, two-way ANOVA and an adequate post-test should be used instead of a t-test or one-way ANOVA. The authors are urged to consult a certified statistician or use appropriate tests to analyze their experiments.
Minor issues.
1 .- Please carefully correct for grammar and standard English usage throughout the text.
2 .- Correct the caption of Figure 2. It should read PKR1.
3 .- Bar graphs are not histograms. Please correct the use of the word "histogram." A histogram is an approximate representation of the distribution of numerical data. A column or bar graph summarizes tendencies and data variance (in the form of SD, SEM, or other variance measures). In this same vein, it is uncertain why the authors chose to show a representative experiment in their bar graphs when they missed the opportunity to summarize all the data at hand.
Overall, with the appropriate controls, the ideas and results presented in this manuscript should interest the field.
Comments on the Quality of English LanguagePlease carefully correct for grammar and standard English usage throughout the text.
Author Response
Reviewer 2
An expanding role for melanocortin accessory proteins (MRAPs) is becoming increasingly evident. The function of these single-membrane-spanning proteins goes beyond the melanocortin system, and the interaction with an increasing list of GPCRs transcends the modulation of constitutive activity or receptor membrane chaperoning. For example, an interplay between arrestin recruitment and MRAP2 (the isoform predominant in the brain and other peripheral tissues outside the suprarenal glands) was described for ghrelin receptors and is the subject of the study presented by Lattanzi and collaborators herein. This manuscript argues that MRAP2 inhibits beta-arrestin-2 (a.k.a. arrestin-3) recruitment by the prokineticin receptor 2. This is a provocative proposition that posits a broader role for MRAPs in general as critical modulators of GPCR function.
To support their results, the authors perform cell-based IHC and BRET-based arrestin recruitment assays in the presence of increasing amounts of transfected MRAP2, pull-down experiments, and ERK phosphorylation as a surrogate of G-protein and arrestin-mediated signaling. At face value, the results presented by the authors seem to support their claim, but the lack of fundamental controls holds back these assertions.
Major concerns
1 .- Protein expression squelching is an artifact perniciously ignored in the literature, significantly leading to erroneous result misinterpretations. The effects of over-expression that ultimately affect overall protein expression should be considered when analyzing data. For example, increasing the concentration of MRAP2 by twenty-fold will have a non-linear (or easy to predict) impact on PKR2 and arrestin cell expression. A recognized limitation of BRET-based experiments is finding the most idoneous way to account for this effect. For example, if available, sometimes a negative or loss-of-function mutant of the protein being studied should be used in lieu of the use of no protein at all, as the relative expression of the other components of the system WILL be affected by the absence or presence of the protein being studied. In this regard, the authors do not show how the presence or absence of MRAP2 affects the expression of PKR2 or Arr. BRET assays are particularly sensitive to this artifact and WILL lead to erroneous data interpretation if not adequately accounted for. Since the truncated form of the PKR2 seems to indicate that arrestin recruitment is driven by the intracellular loops rather than the receptor C-terminal domain, it is unlikely that the truncated receptor could be used as proper control. To account for these artifacts, the authors could resort to careful equalization of fluorescence and luminescence absolute (or raw) readouts (which are directly proportional to protein expression), but most importantly, by demonstrating or quantifying the cell surface expression of PKRs and MRAPs (since this would not be feasible for arresting due to their cytosolic nature) by cell surface ELISAs, flow cytometry, or the appropriate method of their choosing.
We agree with all your observations on the topic however we believe that our cellular system is not affected by this problem for three main reasons.
First: the stable expression of rGFP-barr2 in CHO guarantees the same expression level in all experimental conditions.
Second: the total amount of cDNA Vectors (420ng/cm2) is identical in all conditions. Specifically, the constant amount of receptor encoding Vector (20ng/cm2) is expressed in association with either Empty Vector (EV) (1:0) or MRAP2 plus EV(1:10) or MRAP2 alone (1:20)
Third: the effect of MRAP2 Is dose dependent, suggesting that the effect Is specific
Concerning the effect of MRAP 2.on the membrane trafficking we trust the works published by (23)
Finally, regarding the suggestion about the use of a loss of function mutant, we would point out the results obtained by the use of the deleted form of MRAP2 (131CT-MRAP2). This mutant is unable to affect receptor activity although it is still active in modulating receptor trafficking at the plasma membrane. The observation that 131CT-MRAP2 is ineffective in modulating PKR2-b-arrestin-2 interaction supports the hypothesis that wild type MRAP2 modulate PKR2-b-arrestin-2 interaction beyond interfering with receptor trafficking
2 .- Low-quality blots for pull-down and ERK phosphorylation experiments do not support the author's conclusions and (respectfully) seem detracted from the quantifications. The authors should present better-quality blots in a revised manuscript.
Thank you. We presented better quality blots.
3 .- Inadequate statistical analyses. It stands to remind the authors that the methodology used to test an experimental hypothesis is determined a priori. Suppose an experiment was designed to test the effect of two continuous independent variables on a continuous dependent variable (e.g., ERK phosphorylation). In that case, two-way ANOVA and an adequate post-test should be used instead of a t-test or one-way ANOVA. The authors are urged to consult a certified statistician or use appropriate tests to analyze their experiments.
Thank you. As you suggested we use Two-way ANOVA followed by followed by Sidak's multiple comparisons test.
Minor issues.
1 .- Please carefully correct for grammar and standard English usage throughout the text.
Thank you. We corrected grammar and standard English usage throughout the text.
2 .- Correct the caption of Figure 2. It should read PKR1.
We corrected the caption in Figure 2 with PKR1.
3 .- Bar graphs are not histograms. Please correct the use of the word "histogram." A histogram is an approximate representation of the distribution of numerical data. A column or bar graph summarizes tendencies and data variance (in the form of SD, SEM, or other variance measures). In this same vein, it is uncertain why the authors chose to show a representative experiment in their bar graphs when they missed the opportunity to summarize all the data at hand.
Thank you. We corrected the word "histogram" with “bar graphs”.

Round 2
Reviewer 1 Report
Comments and Suggestions for Authors
Most questions mentioned in the review were addressed by the Authors.
Some issues remain:
The mentioned scheme of interactions in introduction is presented as a scheme. Still, the results of the experiment are described at the end of introduction, but the design of experiments is not explained?
The explanation provided by Authors for question 6 is satisfying, but the text is not changed and a reader will be confused by two time periods.
Please check the sentence: CTMRAP2 contains the region extending from the residue at position 78 to the residue at position 204 and 131CT-MRAP2 mutant containing only the C-terminal region between the residue at position 78 and the residue at position 131.
The red part describes a shorter form, but with lost C-terminal part of the protein (131-204)?? Please verify, as two lines below the C-terminal part is indicated correctly (78-204).
The discussion is improved (please check "basal binding of ô€€€ arrestin-2 to the PKR2" in the first line of the last part).
Comments on the Quality of English Language
In the text, CO2 is used instead of the correct version CO2
Author Response
Some issues remain:
The mentioned scheme of interactions in introduction is presented as a scheme. Still, the results of the experiment are described at the end of introduction, but the design of experiments is not explained?
We thank you. We explained the design of experiments in introduction.
The explanation provided by Authors for question 6 is satisfying, but the text is not changed and a reader will be confused by two time periods.
We thank you. We clarified the choice of the two time periods in the text.
Please check the sentence: CTMRAP2 contains the region extending from the residue at position 78 to the residue at position 204 and 131CT-MRAP2 mutant containing only the C-terminal region between the residue at position 78 and the residue at position 131.
The red part describes a shorter form, but with lost C-terminal part of the protein (131-204)?? Please verify, as two lines below the C-terminal part is indicated correctly (78-204).
We thank you. We have changed the text. We apologise for the error.
The discussion is improved (please check "basal binding of b arrestin-2 to the PKR2" in the first line of the last part).
We thank you. We modified the sentence.

Reviewer 2 Report
Comments and Suggestions for Authors
This reviewer appreciates that many of the initial concerns were addressed in the revised version of the manuscript presented by the authors. However, this reviewer respectfully disagrees with the author's assertion that a stable cell line is impervious to expression squelching. The assumption that plasmid transfection amounts are directly proportional to protein expression is also not correct. Finally, the introduction of “blank” plasmids in the transfection scheme does not solve expression squelching issues since the root of the artifact is not in saturating the capacity to introduce exogenous DNA inside the cell but rather the saturation or overwhelming of the expression machinery with strong-promoter directed gene expression. As stated in the original review, this, unfortunately, is an underappreciated artifact that has percolated across the reported in vitro assays in the literature. If someone incurred the same mistake in the past and it got published, this does not give a license to continue propagating the same mistake. What is most unfortunate is that this is actually very easy to remedy. In BRET assays, raw luminescence and fluorescence values directly reflect the expression levels for the donor and acceptor. Alternatively, properly tagged proteins could be used for cell-surface ELISA experiments (with the inclusion of proper controls) or even fluorescence-based flow cytometry to quantify surface expression. I urge the authors to take the time to implement the necessary controls before publication.
Comments on the Quality of English LanguageEnglish was improved in the resubmitted version of the manuscript, but some minor errors persist.
Author Response
This reviewer appreciates that many of the initial concerns were addressed in the revised version of the manuscript presented by the authors. However, this reviewer respectfully disagrees with the author's assertion that a stable cell line is impervious to expression squelching. The assumption that plasmid transfection amounts are directly proportional to protein expression is also not correct. Finally, the introduction of “blank” plasmids in the transfection scheme does not solve expression squelching issues since the root of the artifact is not in saturating the capacity to introduce exogenous DNA inside the cell but rather the saturation or overwhelming of the expression machinery with strong-promoter directed gene expression. As stated in the original review, this, unfortunately, is an underappreciated artifact that has percolated across the reported in vitro assays in the literature. If someone incurred the same mistake in the past and it got published, this does not give a license to continue propagating the same mistake. What is most unfortunate is that this is actually very easy to remedy. In BRET assays, raw luminescence and fluorescence values directly reflect the expression levels for the donor and acceptor. Alternatively, properly tagged proteins could be used for cell-surface ELISA experiments (with the inclusion of proper controls) or even fluorescence-based flow cytometry to quantify surface expression. I urge the authors to take the time to implement the necessary controls before publication.
We measured the levels of luminescent receptors (either PKRT2wt/rLuc or PKR2-ΔC52-rLuc) and fluorescent β-arrestin-2 in CHO cells (stably expressing rGFP-β-arrestin-2-GFP) after transient transfection with equal amounts of either luminescent receptor and increasing amounts of vectors expressing MRAP2.
Luminescence was measured on total protein preparations (cps/ug proteins) using a plate luminometer, a Victor X3 Multilabel Reader (Perkin Elmer). The hystogram represents the mean/+ SEM of triplicate determination of cps values obtained in each individual experimental condition (receptor/MRAP2 cDNA ratio: 1:0; 1:5; 1:15).
GFP levels were analysed on cell monolayers of transfected cells. The assay was performed in opaque 96-well plates in a Victor X3 Multilabel Reader (Perkin Elmer). The results showed a comparable level of fluorescence in all experimental conditions analysed (A, B).
The data obtained show that the expression of β-arrestin-2-GFP is almost constant and is not affected by the presence of MRAP2. On the contrary, the expression of PKR2 is modulated by the presence of MRAP2. In particular, the reduction of about 45% in the presence of a receptor/MRAP2 ratio of 1:15 confirms the data already available in the literature (Chaly et al., 2016) (C). However, our data emphasise that in the presence of a receptor/MRAP2 ratio of 1:5, although there is no strong reduction in receptor expression (approximately 20%) (C), we observe a drastic decrease in β-arrestin-2 recruitment in the presence of PK2 ligand and a concomitant increase in β-arrestin-2-PKR2 interaction in the absence of PK2 ligand (basal binding). To confirm these data, we performed BRET experiments with PKR1, whose expression is less affected by the presence of MRAP2. We have also used a MRAP2 mutant, which has been shown (Rouault, A.A.et al.; 2017) to affect the localisation of the PKR1 receptor, but we have shown that it alters the binding of β-arrestin-2 to the PKR1 receptor in the presence or absence of the PK2 ligand. MRAP2 modulates the expression of both WT-PKR2 and the PKR2-ΔC52 mutant, so it is possible to compare the effects of MRAP2 on β-arrestin-2 binding to the two receptors (D).
|
|||||||
|
|
||||||
|
|

Round 3
Reviewer 2 Report
Comments and Suggestions for Authors
This reviewer commends the authors for their efforts in determining the effects of increasing amounts of MRAP2 on arrestin or receptor expression. It is not surprising to see that increasing the transfected amounts of MRAP2 does not affect arrestin expression. From this reviewer’s past experience, arrestins are notoriously amenable to overexpression. However, the authors seem to have ignored the effects of squelching on receptor expression. This should be discussed in the manuscript, but more importantly, the results shown in the response should also be included as part of Figure 1 in the manuscript. Far to often, these artefacts are ignored in the reported literature. Beyond the sake of scientific clarity, including the graphs (column graphs, not histograms...) from the response as part of Figure 1 would be of service to the scientific community at large.
Comments on the Quality of English LanguageNothing to add.
Author Response
This reviewer commends the authors for their efforts in determining the effects of increasing amounts of MRAP2 on arrestin or receptor expression. It is not surprising to see that increasing the transfected amounts of MRAP2 does not affect arrestin expression. From this reviewer’s past experience, arrestins are notoriously amenable to overexpression. However, the authors seem to have ignored the effects of squelching on receptor expression. This should be discussed in the manuscript, but more importantly, ar to often, these artefacts are ignored in the reported literature. Beyond the sake of scientific clarity, including the graphs (column graphs, not histograms...) from the response as part of Figure 1 would be of service to the scientific community at large.
We thank you for the suggestion. We included the results shown in the previous response as parts of figure 1 and figure 3 addind, also, a short explanation in the results.